# Transcriptomic Analysis Reveals Molecular Mechanisms Underpinning Mycovirus-Mediated Hypervirulence in *Beauveria bassiana* Infecting *Tenebrio molitor*

**DOI:** 10.3390/jof11010063

**Published:** 2025-01-15

**Authors:** Charalampos Filippou, Robert H. A. Coutts, Ioly Kotta-Loizou, Sam El-Kamand, Alexie Papanicolaou

**Affiliations:** 1Department of Medicine, School of Medicine, European University Cyprus, 2404 Nicosia, Cyprus; 2Department of Clinical, Pharmaceutical and Biological Science, School of Life and Medical Sciences, University of Hertfordshire, Hatfield AL10 9AB, UK; r.coutts@herts.ac.uk (R.H.A.C.); i.kotta-loizou2@herts.ac.uk (I.K.-L.); 3Hawkesbury Institute for the Environment, Western Sydney University, Richmond, NSW 2753, Australia; sam.el-kamand@westernsydney.edu.au; 4Department of Life Sciences, Faculty of Natural Sciences, Imperial College London, London SW7 2AZ, UK

**Keywords:** polymycovirus, mycoviral infection, hypervirulence, transcriptomic profiling, *Tenebrio molitor*, *Beauveria bassiana*, RNA_Seq, gene expression

## Abstract

Mycoviral infection can either be asymptomatic or have marked effects on fungal hosts, influencing them either positively or negatively. To fully understand the effects of mycovirus infection on the fungal host, transcriptomic profiling of four *Beauveria bassiana* isolates, including EABb 92/11-Dm that harbors mycoviruses, was performed 48 h following infection of *Tenebrio molitor* via topical application or injection. Genes that participate in carbohydrate assimilation and transportation, and those essential for fungal survival and oxidative stress tolerance, calcium uptake, and iron uptake, were found to be overexpressed in the virus-infected isolate during the mid-infection stage. Mycotoxin genes encoding bassianolide and oosporein were switched off in all isolates. However, beauvericin, a mycotoxin capable of inducing oxidative stress at the molecular level, was expressed in all four isolates, indicating an important contribution to virulence against *T. molitor*. These observations suggest that detoxification of immune-related (oxidative) defenses and nutrient scouting, as mediated by these genes, occurs in mid-infection during the internal growth phase. Consequently, we observe a symbiotic relationship between mycovirus and fungus that does not afflict the host; on the contrary, it enhances the expression of key genes leading to a mycovirus-mediated hypervirulence effect.

## 1. Introduction

Entomopathogenic fungi such as *Beauveria bassiana* are promising biopesticides for the control of a wide range of insect pests. This is topical as insect pest and vector control programs are currently being threatened by increasing insecticide resistance [1,2,3,4]. In addition to addressing pesticide resistance, the interest in biocontrol agents such as *B. bassiana* is increasingly driven by their alignment with ONE HEALTH principles, which confer significant benefits across human, animal, and environmental health [5,6,7,8]. These biocontrol agents provide an environmentally sustainable alternative to chemical pesticides, thereby reducing ecological chemical burdens and minimizing human exposure to toxic substances [9]. Furthermore, they support the health of beneficial insects and non-target species, thereby preserving biodiversity and maintaining ecosystem balance. As integral components of Integrated Pest Management (IPM) strategies, these agents facilitate sustainable pest control practices that are both economically beneficial and ecologically sound [10].

*Beauveria bassiana* is a saprophytic cosmopolitan entomopathogen that can infect insects through spore adhesion and penetration. Infection by these entomopathogens resembles dimorphism observed in human fungal pathogens such as *Histoplasma capsulatum*, *Blastomyces dermatitidis*, and *Paracoccidioides brasiliensis* [11]. The *B. bassiana* route of infection begins with the adhesion of spores or aerial conidia to the host cuticle via hydrophobin proteins HYD1 and HYD2. Following adhesion, the penetration of the insect cuticle and epicuticle with hyphae occurs via hydrolytic enzymes including chitinases (*chit1* and *chit2*), proteases (PR1 and CDEP1), and cytochrome P450 (Cyp52X1), which are critical for fungal virulence [12]. After reaching the host insect hemocoel, which is rich in nutrients such as trehalose, fungal cells switch to unicellular blastospores, modifying their cell wall structure to evade host immunity in response to hemocyte recognition. This yeast-like morphology allows the fungal blastopores to grow quickly, establish an infection, and accelerate nutrient utilization, resulting in the mummification of the insect host [13]. During colonization of the host, hemocoel *B. bassiana* demonstrates a parasitic behavior that overwhelms insect defense by employing an arsenal of secondary metabolites and toxic molecules (i.e., beauvericin, oosporein, bassianolide, tenellin, and beauverolides) for inhibition of antifungal response, stress resistance, fast assimilation of nutrients, and rapid growth [14]. After insect host death, the fungus continues to secrete antimicrobial compounds, such as oosporein, that suppress competing microbes, allowing it to completely drain the insect of nutrients, and it begins growing from the inside out. This results in complete coverage of the insect with conidia, leading to a white fluffy growth also known as the “white muscardine” disease [15].

*B. bassiana*, together with other entomopathogenic fungi within the phylum Ascomycota and order Hypocreales, offers significant advantages as compared to classic chemical approaches to control insect populations. Some of their advantages include cost effectiveness, rapid infection of their insect hosts, and extreme efficiency in insecticide resistance or integrated pest/vector management programs [6,7,8]. However, the time required to kill insect pests varies markedly between different fungal isolates and can range from days to weeks depending on the strain. This variability may be due to the fungal genome and/or the presence of extra chromosomal elements such as mycoviruses that potentially modulate fungal virulence [16,17].

Mycoviruses are a novel and diverse group and their potential use in biological control has dramatically increased the interest in their genomic organization and infection cycle. Mycovirus genomes comprise double-stranded (ds) RNA, positive- and negative-sense single-stranded (ss) RNA, or rarely ssDNA, but not dsDNA as of yet. Almost exclusively, dsRNA mycoviruses are incapable of extracellular transmission, but they can be transmitted horizontally from one fungal strain to another either via anastomosis, or vertically during the formation of sexual or asexual spores [18]. Mycoviruses are found in all major groups of fungi and the majority cause no effects on their hosts [19]. However, some mycoviruses decrease or increase the virulence of the fungus, resulting in, respectively, hypovirulence or hypervirulence [18]. The most studied example of hypovirulence is that of *Cryphonectria parasitica*, the causal agent of chestnut blight disease, infected by a positive-sense ssRNA virus from the family *Hypoviridae* [20,21].

Hypervirulence has been described for members of the multicomponent mycoviruses in the two families, *Polymycoviridae* and *Chrysoviridae* [16,22,23], which elicit an increase in growth and virulence for their hosts [17,24,25]. An association between mycovirus infection and hypervirulence has been documented for some *B. bassiana* isolates (ATHUM 4946 and EABb 92/11-Dm). Phenotypic alterations include increased radial growth rate, increased sporulation, and sometimes increased pigmentation in *B. bassiana* [25]. To this end, it is imperative to study and document the role of mycoviruses during infection of insect pests by *B. bassiana* to understand the molecular effects of mycoviruses on gene expression prior to their use as biocontrol agents.

In this study, we demonstrate mycovirus-mediated hypervirulence of *B. bassiana* against the larval form of *Tenebrio molitor* (army mealworm beetle) by comparing virus-free (BbVF) and virus-infected (BbVI) isogenic lines derived from the same *B. bassiana* strain, EABb 92/11-Dm, and *B. bassiana* strains ATCC 74040 and GHA, also known as Naturalis and BotaniGard, respectively. By sequencing messenger (m) RNA 3′-termini and a genome comparison approach, we profiled the transcriptome of the four *B. bassiana* isolates to reveal genetic mechanisms underpinning virulence of the four isolates both in vivo and in vitro [26].

Our study demonstrates that mycovirus infection affects a variety of genes involved in carbohydrate transport and metabolism, secondary metabolite secretion, calcium and iron uptake, ATP-binding cassette (ABC), and multidrug transporters, and biotic and abiotic stress tolerance. Overall, these genes are responsible for the expression of key proteins involved in fungus pathogenicity and are generally upregulated in BbVI as compared to BbVF. Finally, throughout this manuscript, it is important to emphasize the distinction between transcript levels and protein production, acknowledging that increased transcripts do not necessarily translate to increased protein levels.

## 2. Materials and Methods

### 2.1. Maintenance of Fungal Isolates

This study relies on a *B. bassiana* isolate named “EABb 92/11-Dm” which is infected with two mycoviruses, *Beauveria bassiana* non-segmented virus (BbNV) 1 and Beauveria bassiana polymycovirus (BbPmV) 1. This strain belongs to the culture collection of the Department of Agronomy of the University of Cordoba (Spain) and was originally isolated from infected specimens of the Moroccan locust *Dociostaurus maroccanus* (Thunberg) in the breeding area of La Serena in Badajoz (Spain). This strain has been deposited in the Spanish Type Culture Collection located at the University of Valencia under accession number CECT 20376 [27]. The virus-infected and virus-free isogenic lines used were generated as reported previously [17] and are herein designated BbVI and BbVF, respectively. Two virus-free, commercial isolates, ATCC 74040 and GHA, were also used. All isolates were maintained on potato dextrose agar (PDA; Sigma-Aldrich, St. Louis, MO, USA) or Sabouraud dextrose agar (SBA; Sigma-Aldrich, St. Louis, MO, USA) at 25 °C and 80% relative humidity [28]. A cocktail of antibiotics (ampicillin, kanamycin, and streptomycin, each at a final concentration of 100 μg/mL) was used during the cultivation of all strains to prevent bacterial contamination. The use of antibiotics ensured that bacterial contaminants did not interfere with the growth and observations of the fungal strains, thereby maintaining the integrity and accuracy of the experimental results.

### 2.2. Infection of Tenebrio molitor with Beauveria bassiana

*Tenebrio molitor* larvae in the final instar stage were obtained from BioSupplies (Bundoora, VIC, Australia) and stored prior to use in wood shavings in the dark to prevent pupation. Within 5 days of delivery, 10 yellow-colored larvae, weighing ~0.2 g each, were randomly chosen for experimentation. All experiments were repeated on two independent occasions. All experiments were conducted using freshly prepared conidiospores taken from 14-day-old fungal cultures by adding 10 mL of double-distilled (dd) water to each PDA plate and using a sterilized scalpel to liberate conidiospores from the media surface. Conidiospores were then filtered through two-layers of sterilized MiraCloth and the flow-through was collected in a sterile funnel and washed twice with dd water following centrifugation at 4000 rpm. The concentration of conidiospores was assessed using a hemocytometer and adjusted to the desired concentration by serial dilution prior to inoculation of *T. molitor* larvae. Fungal conidiospore germination efficiency was determined following transfer and even spreading of the inoculum onto PDA plates using a sterile glass rod. The plates were then incubated for 24 h at 25 °C in complete darkness prior to examination and counting of the germlings. The percentage of the germlings and conidiospore viability was determined and compared to the hemocytometer count. *T. molitor* larvae were injected with 5 μL of a fungal spore suspension at a concentration of 1 × 10^5^ spores/mL (total of 500 conidiospores) into the hemocoel, at the second visible sternite above the legs, using a Hamilton syringe with a 22s-gauge needle (Hamilton, Reno, NV, USA). A dose of 1 × 10^7^ spores/larva resulted in 100% mortality within six days after spray inoculation and 48 h after injection with BbVI, whilst a dose of 1 × 10^4^ spores/larva did not lead to infection (Appendix A). Concentrations of 1 × 10^5^ and 1 × 10^7^ spores/larva were chosen as optimal for larval injection and spraying, respectively, since their intermediate pathogenicity level facilitated the determination of differences in virulence between the different isolates. Prior to injection, the syringe was disinfected with three washes of absolute ethanol and finally with sterile dd water. These washing steps were repeated between injections with different fungal isolates but not for replicates. Between replicates, the syringe was washed with sterile dd water only. Prior to injection, the larvae were anesthetized by placing them on ice for 5 min. Any larvae that were not responsive to touch were rejected for experimentation. For spray inoculation, which resembles the natural route of infection, 1 × 10^7^ spores/mL suspensions were used. Here, insects were completely submerged into PBS containing the conidiospores for 30 to 60 s. Control experiments included untouched larvae, and only-PBS-injected larvae. Following both inoculation procedures, the larvae were put into Petri dishes containing a piece of carrot, changed every 2 days, to maintain normal moisture levels and be habitable for the larvae. The infected larvae were incubated in darkness at 25 °C for 10 days, and mortality was recorded daily together with observations on melanization and lack of motility. Likewise, after larval death, mycosis was recorded daily (Appendix A). Survival curves were plotted using the mean survival of *T. molitor* larvae over an 8-day incubation period. Survival curves were statistically analyzed according to Kaplan–Meier estimation using GraphPad Prism 8.0 software and *p* values were calculated using Welch’s *t*-test in Rstudio (Appendix A). Furthermore, R statistical software (version R 4.0.0; R Foundation for Statistical Computing, Vienna, Austria) was used to estimate a logistic (logit) regression model using the generalized linear model (GLM) function to determine the correct median lethal time (LT_50_) for *T. molitor* larvae following infection with *B. bassiana* (Appendix A) [29,30]. At 48 h post infection, while the insects were still alive, three sample larvae were randomly selected, and total RNA was extracted from each one separately. Thereafter, library construction and Paired-End Illumina sequencing were used to generate sequence data. These experiments were performed in triplicate.

### 2.3. In Vitro Time Course Study

To investigate the effects of the viral infection on EABb 92/11-Dm, mycelia from (virus containing) BbVI and (virus-free) BbVF isogenic lines were grown at 25 °C in darkness. Individual samples were harvested at 4, 7, 10, 14, and 21 days post infection (dpi). Total RNA extraction, library construction and paired end Illumina sequencing were performed, and all experiments were performed in triplicate as per the previous section.

### 2.4. Nucleic Acid Extraction

Total RNA extracts from all samples were produced using the E.Z.N.A Fungal RNA Kit (OMEGA, Doraville, GA, USA). All samples were homogenized with liquid nitrogen using a mortar and pestle. For in vitro studies, fungal mycelia weighing ca. 50–100 mg were homogenized, while for in vivo studies, entire larvae were homogenized. Following homogenization, the powder was mixed rapidly with 500 μL of lysis buffer plus 20 μL of β-mercaptoethanol. Total fungal DNA was purified using the Dneasy Plant Mini Kit (Qiagen, Hilden, Germany) according to the manufacturer’s instructions.

### 2.5. Next-Generation Sequencing Using the 3′mRNA-Seq Library Prep kit

Lexogen’s QuantSeq kit FWD HT (Vienna, Austria) was used to generate libraries from polyadenylated RNA for Illumina sequencing. This approach is unique in that only a part of the mRNA is sequenced, ca. 200–600 bp of the 3′ termini of the transcripts (i.e., usually the 3′ untranslated region; UTR), and no prior poly(A) enrichment or rRNA depletion is required. Library generation was initiated with 1 μg of total RNA and oligo(dT) priming according to the manufacturer’s instructions. The libraries were then subjected to a quality control procedure using a 2100 Bioanalyzer, or QIAxcel and Qubit to measure concentration prior to pooling.

## 3. Bioinformatics Analysis

### Quality Control (QC) of Sequencing Reads

The quality of the sequencing data was assessed using Kraken (for contamination; v2.0.8; Johns Hopkins University, Baltimore, MD, USA) and FASTQC (Quality Control of sequencing data; v0.11.9; Babraham Bioinformatics, Cambridge, UK). Sequencing data were processed for clipping residual adapters and trimming low-quality bases using Trimmomatic (v0.39; Usadel Lab, RWTH Aachen University, Aachen, Germany) [31] via JustPreProcessMyReads (v1.0; https://github.com/genomecuration/justpreprocessmyreads, accessed on 1 October 2020) of the Just Annotate My genome (JAMg) software version 8. A second FastQC step was performed to check the consistency of k-mer distributions and GC content, and to eliminate over-represented sequences. The JAMg software uses Trinity RNASeq (which stands for ribonucleic acid sequencing; v2.11.0, Broad Institute, Cambridge, MA, USA) [32], PASA (Program to Assemble Spliced Alignments; v2.4.1 Broad Institute, Cambridge, MA, USA) GeneMark-ES (Eukaryotic Self-training; v4.64, Georgia Institute of Technology, Atlanta, GA, USA), Augustus (named after a Roman Emperor; v3.3.3, University of Göttingen, Göttingen, Germany), and other tools to deliver a genome annotation and gene prediction as per Fritz et al. 2018 with improvements from Bayega et al. 2020 [33,34]. A description of the process is available on https://github.com/genomecuration/JAMg/wiki/Workflow-overview (accessed on November 2021). After trimming and quality control, the reads were aligned against the fungal public genome *B. bassiana* ARSEF 2860 (ASM28067v1) using Trinity RNASeq. The resulting data were assembled into putative transcripts using the Trinity RNASeq de novo and genome-guided approaches. The latter involved using a program called GSNAP (which stands for Genomic Short-read Nucleotide Alignment Program; v2020-10-29, University of California, USA [35]) to align short-read RNA-Seq—from this project and public repositories—to each target genome to obtain spliced reads and junction gap information.

The resulting data and public mRNA sequences were interrogated with PASA (Program to Assemble Spliced Alignments) [32] to define a subset of high-quality full-length transcripts for training Hidden Markov Model prediction software (“PASA GOLDEN”; v3.3.1 in 2020, Howard Hughes Medical Institute, Ashburn, VA, USA). This involved GeneMark-ES [36] and Augustus [37], gene prediction tools using exon–intron junction information from the previous steps. Augustus used further external evidence from the JAMg pipeline and public data (such as transcriptome sequencing, annotations of closely related genomes, or repeat information) to improve precision and completeness of the automated annotation. A penalty of 0.64 was set for intron predictions that were not supported by transcriptome data.

These results were reintegrated in the PASA database to create a non-redundant set, identify alternative transcription, and UnTranslated Regions (UTRs). The results of this database were integrated with protein alignments, the Augustus, and GeneMark-ES results to deliver a final Official Gene Set (OGS) for each genome. The results were then visualized with the Apollo plugin (Apollo v2.6.1, Berkeley Bioinformatics Open Projects, Berkeley, CA, USA of JBrowse (JavaScript Browser; v1.16.11, Comparative Genomics Lab, Indiana University, Bloomington, IN, USA) [38] and manually corrected (“curated”) using our biological expertise.

The gene expression analysis was qualitative using DEW (Differential Exppression on the Web; Papanicolaou A https://github.com/alpapan/DEW/wiki commit 3fb77e5, accessed on 1 October 2020), which aligns the 3′UTR reads to the mRNA OGS using GSNAP, accounts for isoforms using salmon to generate effective counts, normalizes by library size, biological co-efficient, and gene length to deliver a TMM (Trimmed Mean of M-values) of TPM (Transcripts Per Kilobase Million) values and informative plots for each gene.

## 4. Results and Discussion

### 4.1. Tenebrio Molitor Survival Assay Following Beauveria bassiana Infection

The army mealworm, *T. molitor*, belongs to the order *Coleoptera* within the cohort *Endopterygota* (also known as *Holometabola*, which undergo transformation from larvae to winged adults via pupation) and was used to compare the virulence of isogenic lines BbVI, BbVF, Naturalis, and BotaniGard. As different *B. bassiana* strains have different growth and sporulation rates, which result in differences in virulence, preliminary experiments were performed to determine the optimal spore concentrations required to elicit clear differences in pathogenicity between the four *B. bassiana* isolates. To this end, mealworm larvae were injected or sprayed with serially diluted spore suspensions ranging in concentration from 1 × 10^4^ to 1 × 10^7^ spores.

Kaplan–Meier survival analysis showed that there were statistically significant differences (*p*-value < 0.05) of increased mortality for BbVI as compared to the other three isolates following inoculations using either technique (Appendix A). Topical application of BbVI demonstrated an LT_50_ of 4.5 days as compared to its isogenic line BbVF, Naturalis, and BotaniGard, which had an LT_50_ of, respectively, 5.7, 5.1, and 5.3 days (Appendix A). The differences in LT_50_ values between BbVI and the other isolates exhibited a span of ca. 24 h, illustrating significant mycovirus-associated hypervirulence. Similar results were obtained using larval injection, where BbVI demonstrated an LT_50_ of 3.5 days, while BbVF, Naturalis, and BotaniGard had an LT_50_ of, respectively, 4.7, 3.8 and 3.9 days (Appendix A). Taken together, there were no major differences in the trends of the isolates investigated between injection or spray inoculation.

These results indicate major differences in virulence between BbVI and BbVF, with the virus-infected isolate BbVI eliciting accelerated hypervirulence within 24 h post inoculation. This time period corresponds to the time period required for *B. bassiana* to penetrate the insect cuticle and establish infection [39] and is congruent with previous observations concerning *B. bassiana* and its virulence for other members of the *Endopterygota* cohort *viz. Galleria mellonella* (order *Lepidoptera*) and *Ceratitis capitata* (order *Diptera*) [16,17].

### 4.2. Genome Structural Annotation of Beauveria bassiana Isolates

All genomic data were examined for evidence of contamination. For the BbVI and BbVF isolates, which were grown on PDA in a time course, alignment of the RNA-seq reads, each 152 base pairs in length, with the *B. bassiana* (ASM28067) genome using Bowtie2 (v2.4.2; Langmead Lab Johns Hopkins University, Baltimore, MD, USA) confirmed that >90% of the reads were derived from *B. bassiana*. Similarly, alignment of the reads with the *T. molitor* genome revealed that >90% of the reads were derived from the insect while the remaining reads mostly aligned with *B. bassiana,* with some contaminants from, e.g., *Aspergillus flavus,* possibly reflecting conserved regions.

Several reads from BbVI taken 21 days post inoculation (dpi) aligned with a sequence identified as Beauveria bassiana RNA non-segmented virus 1 (BbNV-1) first isolated from *B. bassiana* isolate EABb 92/11-Dm [40]. Although this B. bassiana isolate is also infected with Beauveria bassiana polymycovirus 1 (BbPmV-1), no reads matching its sequence were recorded. This lack of detection is due to polymycoviruses not having a poly(A) tail.

In this study, we used 3′ Lexogen gene expression quantification and relied on well-characterized UTRs in order to assign the reads correctly and provide an accurate individual gene expression profile (Appendix A). However, neither *B. bassiana* nor *T. molitor* species are model organisms or have large genomics communities to provide public annotations of sufficient quality. Therefore, we conducted manual curation for all genes of interest, adding information (metadata) and corrections to eventually provide a high-quality dataset for eleven *B. bassiana* genomes. More specifically, genes of interest were manually curated to detect start and stop codons, UTRs, introns and exons, and alternative splicing.

Alternative splicing is an important cellular process which allows multiple proteins to be expressed from a single gene [41]. Following manual curation, an alternatively spliced isoform was found for the *atg1* gene, an α-glucose transporter, responsible for fungal virulence, host colonization, germination and conidial yield (Appendix A) [42]. The exact function of the alternatively spliced *atg1* transcript is unknown, but may be similar to that of the original *atg1*, which interacts with other genes involved in protein targeting and autophagy regulation, impacting various cellular pathways [43]. Alternatively, spliced isoforms were found for many other genes during manual curation (Appendix A).

Since genome annotation provides the first insight into each organisms’ gene collection, correct annotation of transcript variants and their encoded proteins is essential to produce a complete catalogue of predicted transcripts. One issue encountered during manual curation was that some genes appeared as single exons but RNA-seq coverage plus intron/exon junction reads evidence proved otherwise. The *cdep1* gene, a subtilisin-like protease, responsible for cuticle degradation, hyphal extrusion and conidiation [44], was expected to comprise a single exon based on homology predictions. However, based on RNA-seq coverage and junction reads, the *cdep1* gene comprises four additional exons and three introns. This gene was curated following the addition of its UTRs (Appendix A), and cross-referenced with known proteins, and the “OGS”, “PASA Golden”, and “GeneMark” datasets. To further improve the quality of our data, the 3′-UTRs were subjected to further investigation by crosschecking them against the read alignments (Appendix A). The protocol used for library preparation results in segments up to 600 bases in length including the 3′-UTR, sequenced in the reverse orientation, and the read alignment indicates that the data are representative for this specific gene. Spurious alignments were also an issue (Appendix A). GSNAP alignment allowed for large gaps of up to 70 kb downstream of the 5′-UTR (Appendix A), which were incorrect and could not be used for statistical analysis. Therefore, by combining different methodologies and human oversight we were able to create a more complete description of the transcriptome for all eleven known *B. bassiana* isolates, which is now available.

In order to resolve which reads originated from which gene, both the 3′-UTR direction and the orientation of the reads were assessed to ensure they were opposite to the direction of the gene. Appendix A illustrate an example of Lexogen-derived reads that did not match gene annotations before curation (Appendix A) and after the curation process (Appendix A), demonstrating how the manual curation improved the accuracy of gene annotations.

### 4.3. Mycovirus-Mediated Modulation of Hypervirulence-Associated Genes

Virulence genes are vital for pathogenicity and any reduction in their transcript and subsequent protein production affects infection efficiency. For *B. bassiana*, it is important to consider the activity of specific virulence factors generated by gene families that contribute to insect infection. This requires both comprehensive ascertainment of such gene families and assessment of the expression levels for genes involved in virulence or metabolic pathways. Here, we investigated key pathways involved in fungal virulence by complementing the often misleading *p*/q-values produced by averaging data with high biological variance [45] with boxplots that expert curators can use to qualitatively assess gene levels for each candidate gene.

Each such boxplot represents the normalized (trimmed mean of M values (TMM) and library size) expression profiles of all genes in a particular library and denotes (red dot) the expression profile of a specific gene for a specific sample to allow for a comparison of its expression simultaneously between and within libraries. For zero expression values, the red dot appears at the far left of the X axis. Both traditional F tests (after correcting for multiple testing) and Mann–Whitney tests were conducted to assess the significance of any pairwise expression differences or ranks, respectively, between libraries. However, we found that statistical *p*-values alone were insufficient for prioritizing candidates for downstream work. They fail to account for metadata, the variability among replicate samples, and the potential biological origin of this variability. Consequently, important details may be overlooked. Therefore, as in other parts of this work, we complemented the statistical approaches with an expert-curated qualitative analysis of the underlying data.

For an insect pathogenic fungus to cause disease, it must successfully achieve several stages of pathogenesis: (I) contact, recognition, and adhesion; (II) germination and penetration (colonization requires penetration, as spores rely on endogenous nutrients in this stage); (III) immune response evasion, and infection (secondary metabolites are certainly important in this stage but can also be involved in other stages); (IV) colonization and saprophytic growth (occurring after the insect’s death; Figure 1). During this process, insect pathogenic fungi face several additional stress factors such as temperature, UV-B radiation, and the insects’ first line of defense. Therefore, successful insect infection is not guaranteed and activation of several genes responsible for successful completion of each step is required.

### 4.4. Carbohydrate Assimilation and Transport

Trehalose is the primary carbohydrate found in insect hemolymph, playing a critical role in buffering against osmotic shock in various organisms. *B. bassiana* relies on trehalose for effective growth within the insect hemocoel, facilitated by *agt1*, a gene encoding an α-glucoside transporter crucial for trehalose uptake. Previous studies demonstrated that the deletion of *agt1* leads to diminished growth on diverse carbohydrate substrates, decreased conidial production, and reduced virulence in insects, whether applied topically or injected into the hemocoel [42]. Disruption of the *jen1* gene, which impairs carboxylate transport, similarly affects conidial yield, emphasizing its role in nutrient assimilation [46]. Likewise, the importance of the ATP-binding cassette (ABC) transporter family, including MDR1, MRP1, PDR1, PDR2, PDR5, and PMD1, is evident in their role in multidrug resistance and transport of compounds across the plasma membrane via ATP hydrolysis [47]. Deletion studies of these transporters, as reported by other research groups, show increased susceptibility to fungicides, indicating their protective role against toxic compounds [12,46].

We examined the gene expression levels of various transporters, including *agt1*, *jen1*, and ABC transporters *(mdr1*, *mrp1*, *pmd1*), initially in vitro (Figure 2) and subsequently in vivo. In vitro, all genes were expressed in both BbVI and BbVF at 7–10 dpi, as evidenced by the dot position on the quartiles (Appendix A). The *agt1* gene was moderately to highly expressed and upregulated in BbVI compared to BbVF at 7 and 21 dpi; notably, it was switched off in BbVF at 21 dpi. The same levels of expression for the *jen1* gene were noted at 10 dpi, while at 14 dpi, the *jen1* gene was switched off in BbVF. The *mdr1* gene was highly expressed and upregulated in BbVI compared to BbVF at 7 dpi, while the same levels of expression for the *mrp1* gene were noted at 7 dpi (Table 1).

In vivo, there were notable differences in transporter gene expression between BbVI and BbVF. During infection of *T. molitor* via injection, specific patterns of gene expression were observed. The genes *jen1* and *mdr1* were switched off in all isolates, while *agt1* was expressed only in BbVI, indicating a unique role for carbohydrate transport in this particular isolate following injection (Appendix A). The *mrp1* gene was expressed in all isolates except BbVF, while the ion transport and/or toxin secretion gene *pmd1* was expressed only in BbVI and Naturalis. The relative upregulation of *mrp1* and *pmd1* in BbVI compared to BbVF after injection underscores the enhanced role of these genes in BbVI’s adaptive and virulence strategies. In contrast, when *T. molitor* was infected via spraying, a different gene expression pattern was observed, with all aforementioned genes—*jen1*, *mdr1*, *agt1*, *mrp1*, and *pmd1*—switched off in all isolates (Table 2).

In this study, the expression of transporter genes in vitro and in vivo underscores their significance in boosting virulence and resistance to environmental stresses. We have observed that all transporter genes are upregulated in BbVI as compared to BbVF, suggesting a critical role in sustained nutrient acquisition, especially during the middle to late stages of infection (Figure 1). This enhanced expression may also be due to the fungus secreting toxins to establish a niche, compete for space, and/or evade host immune responses during the middle to late phases following injection, contrasting with potentially less beneficial expression early after spray inoculation, when *B. bassiana* does not need to complete the initial steps of pathogenesis such as penetration. Furthermore, *B. bassiana* isolates naturally infected with either BbPmV-1/BbNV-1 (designated as BbVI in the current study) or another mycovirus, Beauveria bassiana polymycovirus (BbPmV) 3, exhibit faster growth on various carbohydrates (including minimal Czapek-Dox media with trehalose, lactose, glucose, and glycerol, but not maltose and fructose, as a carbon source) compared to their virus-free isogenic lines [25]. The increased growth in the virus-infected isogenic lines is likely due to the upregulation of *agt1*, observed here in BbVI as compared to BbVF, enabling enhanced trehalose uptake and utilization.

The *hxtA* gene, encoding a high-affinity hexose transporter, is induced under glucose starvation and is regulated by the product of the *rco3* gene, which acts as a glucose sensor [48]. This coordinated regulation ensures an efficient uptake of monosaccharides during carbon starvation, which are then metabolized within the mycelia by alpha-1,3-glucanase enzymes.

We examined the gene expression levels of the high-affinity glucose transport and sensory system in vivo, and we noted expression ranging from low to high in all isolates except BbVF. Following injection, the *hxtA* gene was expressed in only two isolates, BbVI and Naturalis, suggesting a response to lower glucose availability. This interpretation is supported by the common regulatory behavior of high-affinity transport systems, which are typically upregulated under nutrient-limited conditions to optimize resource uptake. The *rco3* gene was only expressed in BbVI and Naturalis, showing a slight overexpression in Naturalis based on the dot position in Q4. Additionally, the alpha-1,3-glucanase gene was switched on in BbVI, Naturalis, and BotaniGard, showing similar moderate to high expression levels in both BbVI and Naturalis with the dot position in Q3 and low expression in BotaniGard with the dot position in Q1 (Appendix A). Conversely, following spraying, all genes involved in the high-affinity glucose transport and sensory system were switched off in all isolates, except for low *hxtA* expression in BbVI (Table 2). Additionally, glucose transport activity was apparently influenced by the glucose transporter encoded by the *rco3* gene, which acts as a glucose sensor and may regulate expression of genes for both high- and low-affinity glucose transport [49].

Our observations show that *rco3* and *hxtA* exhibit similar expression patterns. This could indicate that the expression of the hexose transporter *hxtA* may depend on the glucose transporter *rco3*, as suggested by previous models or experimental evidence [50]. Furthermore, the expression profile for the alpha-1,3-glucanase gene showed a similar pattern to *rco3* and *hxtA*, suggesting that the glucose transporter acting as a glucose sensor induces the hexose transporter and alpha-1,3-glucanase under conditions such as carbon starvation. However, it is also plausible that all genes are co-regulated by the same upstream regulatory elements or environmental cues, leading to concurrent expression profiles. This co-regulation could be particularly significant in the context of glucose utilization and depletion, whereupon alpha-1,3-glucanases from the glycoside hydrolase family are secreted to release monosaccharides following glucose exhaustion.

### 4.5. Calcium and Iron Binding Proteins

The calmodulin (CaM) pathway is one of the best characterized: Ca^2+^ binds to CaM and activates various downstream targets, including calreticulin encoded by the *crt* gene, a calcium-transporting ATPase encoded by the *pmr1* gene and various vacuolar calcium transporters including that encoded by the *vcx1* gene (Figure 3) [43]. Calreticulin is a unique endoplasmic reticulum (ER) luminal resident protein which performs two major functions: chaperoning and regulating Ca^2+^ homoeostasis. As a chaperone, calreticulin participates in the synthesis of several molecules, including ion channels, surface receptors, integrins and transporters [51]. The calreticulin/calnexin complex is involved in the folding of newly synthesized glycoproteins that enter the ER before they are exported to various subcellular compartments [52]. Calreticulin also affects intracellular Ca^2+^ homoeostasis by modulation of ER Ca^2+^ storage and transport [51]. The *pmr1* gene encodes a P-type Ca^2+^ ATPase located near the Golgi apparatus, which transfers Ca^2+^ from the vacuole to the Golgi apparatus. P-type calcium ATPase pumps are crucial for sustaining intracellular Ca^2+^ homeostasis and removing excess Ca^2+^ from the cytosol [53,54]. The *vcx1* gene also plays a role in intracellular Ca^2+^ homeostasis. Finally, the *csa1* gene encodes a calcium-binding protein similar to CaM and is linked to the production of organic acids such as oxalate, citrate, lactate, and formate, and the regulation of plasma membrane proton pumps. Organic acids, particularly oxalic acid, are involved in insect cuticle penetration but are not required afterward [12,55].

The expression profile of genes involved in calcium assimilation was investigated in vivo (Table 3). Following injection, the *crt* gene was expressed in all isolates except BbVF and showed upregulation in BbVI compared to BbVF (Appendix A). The *pmr1* gene exhibited low expression levels only in BbVI and the *vcx1* gene was switched off in all isolates (Appendix A). The *csa1* gene was also switched off in all isolates (Appendix A). Following spraying, all genes were switched off in all isolates (Appendix A). This consistent low expression of almost all calcium binding proteins suggests that cytosolic Ca^2+^ concentrations remained within normal ranges.

The known expression patterns of the *csa1* gene, suggesting that once the fungus has penetrated the exoskeleton and initiated infection, the *csa1* gene becomes dispensable [15], correlate with our findings. Following the breach of the insect cuticle by the fungus, the necessity for organic acids diminishes and the organism reallocates its energy and resources towards other processes essential for post-penetration activities [56]. Injection likely provides a more direct and abundant nutrient source, supporting higher metabolic activity and growth, hence the upregulation of specific genes such as *crt*.

The concurrent inactivation of *vcx1* across all isolates suggests that the intracellular Ca^2+^ concentrations are within normal ranges during infection, thus negating the need for *vcx1* expression. In previous studies, the deletion of the *vcx1* gene drastically upregulated the *pmr1* gene, indicating a compensatory mechanism to maintain calcium homeostasis [57,58]. However, in our study, we did not observe a consistent upregulation of *pmr1* across all isolates when *vcx1* was not expressed. This discrepancy could be due to differences between gene deletion and gene downregulation: while a complete deletion triggers a robust compensatory response, the regulatory downregulation of *vcx1* under specific conditions may not elicit the same level of *pmr1* upregulation. Additionally, other calcium transporters or regulatory mechanisms might compensate for the lack of *vcx1* expression, minimizing the need for *pmr1* upregulation.

Iron metabolism in *B. bassiana* involves the regulation of iron uptake, storage, and detoxification to maintain cellular function and prevent toxicity. Ferricrocin is an intracellular siderophore essential for maintaining cellular iron homeostasis and mitigating oxidative stress [59,60]. Disruption of ferricrocin synthesis leads to alterations in fungal development, enhanced mitochondrial activity, and increased expression of genes involved in oxidative stress response and iron homeostasis, which collectively contribute to the fungus’s ability to thrive under iron-limited and iron-rich conditions, increasing its virulence against insect hosts [61]. A set of three *fet3* genes (*fet3a*, *fet3b*, and *fet3c*) encode iron transport multicopper ferroxidases critical for iron acquisition. The expression of these genes is induced under low-iron conditions and during infection, while the disruption of these genes results in impaired fungal growth, reduced virulence, and decreased intracellular accumulation of ferrous iron [62,63,64]. The *mfs* gene belongs to the major facilitator superfamily (MFS), whose members are involved in essential functions such as fungal toxin transportation and act as siderophore transporters in the absence of extracellular Fe^2+^ siderophores [47,65].

The expression profile of genes involved in iron assimilation was investigated in vivo (Table 3). Following spraying, the *fet3* gene was switched off; following injection, *fet3* was expressed in all isolates except BbVF, showing similar high expression levels in both BbVI and Naturalis with the dot position in Q4 and moderate low expression in BotaniGard with the dot position in Q2 (Appendix A). This observation suggests a low-Fe^2+^ environment during late infection following injection but not during mid-infection following spraying. Usually, viruses cause cellular Fe^2+^ overload since essential processes such as genome replication and protein synthesis require Fe^2+^ [66]. BbVI is under Fe^2+^ starvation; however, the mycoviral load does not affect fungal pathogenicity. The *mfs* gene showed consistent high expression with no significant changes across all isolates following either spraying or injection (Appendix A). The consistent high expression of *mfs* across all isolates and conditions may indicate its essential role in maintaining basic metabolic functions, possibly related to siderophore transport or toxin export, independent of external iron levels. On the other hand, the induction of *fet3* expression following injection suggests that the fungus is experiencing iron limitation during late infection stages, triggering specific iron acquisition mechanisms. This further supports the notion that *B. bassiana* adapts to iron scarcity during infection, maintaining essential metabolic functions under varying environmental conditions.

### 4.6. Secondary Metabolite Biosynthesis

Mycotoxins, which are secondary metabolites produced by certain pathogenic fungi, play a crucial role after *B. bassiana* enters the insect hemocoel. To successfully establish infection, *B. bassiana* must evade and/or suppress the host’s immune defenses and inhibit the growth of competing microorganisms. To this end, *B. bassiana* produces and releases various secondary metabolite mycotoxins, such as beauvericin, bassianolide, oosporein, and tenellin, many of which are key contributors to its pathogenicity [67].

The *beas* gene encodes a non-ribosomal peptide synthetase that facilitates the production of beauvericin, a cyclo-oligomer depsipeptide. Beauvericin is toxic to a variety of insect pests, the brine shrimp (*Artemia salina*) [68], Gram-positive bacteria, and several filamentous fungi [69]. The genes *bsls* and *osp1* are responsible for the production of bassianolide, a cyclo-oligomer depsipeptide synthesized by a non-ribosomal peptide synthetase (NRPS), and oosporein, which is synthesized by a polyketide synthase (PKS) encoded by the *osp1* gene as part of a gene cluster critical for the initial step in oosporein biosynthesis [70].

The *tens* gene encodes a fused type I polyketide synthase-nonribosomal peptide synthetase (PKS-NRPS) whose structure contains a 2-pyridone residue and is responsible for the production of the yellow pigment tenellin [71]. Tenellin does not exhibit toxicity towards *G. mellonella* larvae [71] and serves as an iron chelator, capturing iron through its metal-binding, hydroxamic acid group [72,73] and protecting against iron-induced oxidative stress [60]. Particularly under conditions rich in iron, an iron–tenellin complex accumulates in certain mutant strains, while only the intracellular siderophore ferricrocin is present in the wild-type strains [60]. Shortage of ferricrocin triggers the production of tenellin, which then acts as an iron chelator, protecting the cell from oxidative stress caused by elevated iron levels [60].

Additionally, some *B. bassiana* isolates can encode proteins possessing bacterial enterotoxin-like domains that are homologous to enterotoxins secreted by *Beauveria*, *Cordyceps*, and *Metarhizium* species [74]. HC-toxin, a cyclic tetrapeptide produced by the plant pathogenic filamentous ascomycete *Cochliobolus carbonum*, inhibits histone deacetylases in many organisms, including insects and mammals [75].

The expression profile of genes responsible for mycotoxin production in *B. bassiana*, including *beas*, *bsls*, *osp1* and *tens*, was investigated in vitro (Table 4). The *beas* gene, responsible for beauvericin production, showed low expression at all measured timepoints in BbVI, whereas it was switched off at 4 and 14 dpi in BbVF (Appendix A). Contrary to the *beas* gene, the *tens* gene responsible for tenellin production exhibited medium–low to medium–high levels of expression across various timepoints in BbVI. At 21 dpi in particular, *tens* expression was medium–high in BbVI, whereas in BbVF, it was switched off (Appendix A). The genes *bsls* and *osp1*, responsible for the production of bassianolide and oosporein, respectively, were both switched off at all timepoints sampled in vitro (Table 4) and are not essential for pathogenicity when the fungus is grown in a controlled environment.

The expression profile of the aforementioned genes responsible for mycotoxin production, together with the *bba* and *tox2* genes that encode, respectively, a heat-labile enterotoxin and an HC-toxin, was further investigated in vivo. The *beas* gene was expressed at medium to high levels in BbVI following injection but not spraying, was expressed at low levels in BotaniGard and Naturalis both after injection and spraying, and was switched off in BbVF (Figure 4 and Appendix A). Following *T. molitor* infection, high expression of the *tens* gene was observed in all isolates with no major differences between them (Table 5). Notably, tenellin expression levels were higher following spraying as compared to injection (Appendix A). These results might reflect the time of infection: following spraying, the fungus is at the early stages of infection in an iron-replete environment, leading to high tenellin expression to prevent iron-induced oxidative stress. Conversely, during injection, which is considered an early to middle infection stage, tenellin expression is reduced [60,71]. The fungus uses siderophores like ferricrocin to manage iron levels. However, when ferricrocin is deficient, tenellin acts as an iron chelator to prevent harmful ROS through the Fenton reaction. This adaptive response protects the fungus from oxidative stress and maintains iron homeostasis. Thus, elevated tenellin expression following spraying likely responds to the early infection stages, where ferricrocin alone might not manage excess iron effectively (Figure 4) [60]. Similarly to our in vitro studies, both the *bsls* and *osp1* genes were switched off in all four isolates in vivo (Appendix A). Fan and colleagues demonstrated that oosporein is not involved in the early to mid-infection stage but is more likely to function following insect death to inhibit additional microbes, thereby allowing the fungus to fully utilize host nutrients and complete its life cycle [15]. Our sample processing was conducted at 48 hpi, aligning with the mid- to early-late infection stages, during which the insect host remained alive (Figure 1). This timing might explain the inactivation of the *bsls* and *osp1* genes. Finally, the *bba* gene was found to be expressed at low to medium levels only in Naturalis following injection, while it was switched off in all four isolates following spraying. The *tox2* gene was expressed also at low to medium levels in BbVI following either injection or spraying and in Naturalis following injection (Appendix A).

Mechanisms of toxin pathogenicity vary with the type and amount of toxin and the host. For instance, a toxin might be useful for fungal virulence in one host but not in another. In this investigation, the beauvericin biosynthetic gene (*beas*) is switched on in all isolates both in vivo and in vitro, whereas the bassianolide biosynthetic gene (*bsls*) is switched off. These observations may indicate an important contribution of beauvericin to *B. bassiana* virulence against *T. molitor*, while bassianolide may not be essential. Therefore, it is impossible to generalize the strategy of mycotoxin production and virulence during infection [76].

### 4.7. Biotic and Abiotic Stress Tolerance

Almost all organisms are exposed to various stress factors, and entomopathogenic fungi are no exception. These fungi face significant abiotic stresses such as osmotic stress, temperature fluctuations, and UV irradiation exposure. These stressors often lead to the generation of ROS, inducing oxidative stress that can be lethal to the fungi. Catalase, encoded by the *cat* gene, catalyzes the conversion of hydrogen peroxide into water and oxygen, while superoxide dismutase (SOD), encoded by the *sod* gene, catalyzes the dismutation of the superoxide radical (O_2_⁻) into oxygen and hydrogen peroxide. By converting these ROS into less harmful molecules, SOD plays a critical role in preventing cellular damage and supporting the fungus’s ability to survive under oxidative stress. Thioredoxins, encoded by the *trx* genes, are oxidoreductase enzymes that assist with redox balance and function as antioxidants by catalyzing protein reduction via cysteine thiol–disulfide exchange [77].

Moreover, these abiotic stress factors can hinder the fungi’s ability to infect insects effectively. Additionally, infected insects exhibit behavioral fever, where they regulate their body temperature through sun exposure, acting as a defense mechanism to eliminate or diminish the infecting microbe’s capacity to proliferate. Heat shock proteins (HSPs) serve as molecular chaperones, playing critical roles in various physiological processes in fungi and other eukaryotes. Stress-induced HSPs such as HSP1, HSP10, HSP30, HSP70, and HSP90 are essential for maintaining normal fungal functions under stress conditions [78,79]. Specifically, HSP70 is vital for growth, conidiation, pathogenicity, and maintaining cell wall integrity [80], whereas HSP30 is linked to conidial thermotolerance [81]. HSP90 is involved in activating the mitogen-activated protein kinase (MAPK) pathway through the induction of the Hsf1 transcription factor and mediates the communication between the MAPK Slt2 and Hog1 pathways [82,83]. Attempts to create a Δ*hsp90* mutant strain in *Aspergillus fumigatus* by Lamoth and colleagues underscored HSP90′s critical role in fungal survival [79].

Likewise, mannitol and trehalose are significant for fungal spore viability, germination, and stress tolerance. The enzymes mannitol-1-phosphate dehydrogenase and mannitol dehydrogenase, encoded by the *mpd* and *mtd* genes, respectively, are involved in mannitol biosynthesis and are crucial for the full virulence of *B. bassiana* [76]. While these compounds are important in carbohydrate metabolism, their role extends beyond nutrient acquisition. Mannitol plays a crucial role in managing the cytoplasmic pH and also acts as an antioxidant, safeguarding fungal cells from ROS produced during extensive fungal growth [84,85,86]. Trehalose is pivotal not just for carbon metabolism but also for the cellular response to abiotic stresses [87]. The enzymes trehalose-6-phosphate synthase and trehalose-6-phosphatase, encoded by the *tps1* and *tps2* genes, respectively, are involved in trehalose biosynthesis. The *agdC* gene also participates in carbohydrate metabolism and encodes alpha/beta glycosidase, which cleaves the alpha (1,4) glycosidic bond of disaccharides such as maltose to glucose [88], linking carbohydrate metabolism to stress tolerance mechanisms.

The expression profile of genes encoding HSPs (*hsp30*, *hsp70* and *hsp90*), enzymes responsible for biosynthesis of mannitol (*mtd* and *mpd*) and trehalose (*tps1* and *tps2*), and carbohydrate metabolism (*agdC*) in *B. bassiana* was investigated in vitro (Table 6). The *hsp30* and *hsp90* genes were highly expressed in both BbVI and BbVF, while *hsp70* showed increased expression in BbVI as compared to BbVF between 14 and 21 dpi (Appendix A). Gene expression analysis showed high expression of *mtd* and *mpd* across all time points in both BbVI and BbVF. Similarly, *tps1* expression analysis revealed no significant difference between the two isolates. In contrast, *tps2* exhibited upregulation in BbVI as compared to BbVF at 7 and 14 dpi (Appendix A). Finally, the expression levels of *agdC* were upregulated in BbVF as compared to BbVI at 4, 7, and 21 dpi (Table 6).

These expression patterns highlight the adaptive response of the fungus to stress conditions. A comparison of the expression levels of *tps1* and *tps2* with those of *mpd* and *mtd* in vitro revealed a higher expression of the latter, underscoring the critical role of mannitol in enhancing stress tolerance within the fungal cells. These results imply that, while both trehalose and mannitol might contribute to stress responses, mannitol could potentially play a more significant role in the fungus’s ability to withstand stressful conditions [76,89]. Additionally, the mycovirus-mediated upregulation of the *agdC* gene correlates with our previous studies illustrating that mycovirus infection affects the metabolic pathway prior to the conversion of maltose and fructose to glucose [25].

The expression profiles of genes encoding catalase, superoxide dismutase, and thioredoxin, together with the aforementioned genes encoding heat shock proteins (including *hsp1/sti1* and *hsp10*), and enzymes responsible for mannitol and trehalose biosynthesis, were further investigated in vivo (Table 7). Following spraying, the vast majority of these genes were switched off in all isolates, with the exception of *trx* in BotaniGard (Appendix A) and *hsp70* in BbVI (Appendix A), both expressed at low to medium levels. Following injection, several genes were expressed at high levels. Among the antioxidants, the *cat* gene showed high expression in BbVI and Naturalis, the *sod* gene was expressed exclusively in BbVI and the *trx* gene was expressed in all isolates except BbVF (Appendix A). Overall, high expression levels of *hsp1/sti1*, *hsp10*, *hsp70*, and *hsp90*, but not *hsp30*, were noted in BbVI and Naturalis. Only *hsp70* and *hsp90*, both of which have broad stress response roles, were expressed at low to medium levels in BotaniGard, while all genes were switched off in BbVF (Appendix A). The lower expression of *hsp30* may be due to its specific role in managing plasma membrane stability and energy metabolism under severe stress conditions, which were likely not present. Conversely, *mpd* and *tps1* were moderately expressed only in BbVI, while *mtd* and *tps2* were switched off in all isolates following all insect infections (Appendix A). Both gene pairs (*mpd/mtd* and *tps1/tps2*) are involved in the biosynthesis of mannitol and trehalose, respectively, but they are not entirely interchangeable. The presence of both gene types is generally required for the efficient synthesis of these compounds, with *mpd/mtd* contributing to different steps in mannitol biosynthesis, and *tps1/tps2* playing distinct roles in the production of trehalose.

It is plausible that, during late infection, the fungus experiences considerable environmental stress, primarily due to nutrient depletion or the accumulation of toxic byproducts. This stress can significantly impact the fungus’s ability to thrive, adapt, and complete its life cycle effectively [90]. This differential gene expression suggests a strategic adjustment by the fungus to the host environment, possibly to optimize survival and virulence by mitigating oxidative stress and maintaining internal pH stability [90,91]. These results suggest that virus infection might influence the expression of these genes, potentially as part of the fungus’s response to stress or to enhance its virulence.

Among the antioxidants, the high expression of *cat* in BbVI and Naturalis indicates a potential reliance on peroxide detoxification in certain isolates, which might be crucial for their survival and pathogenicity. The exclusive expression of *sod* in BbVI suggests a unique role for superoxide dismutase in its defense mechanism, possibly due to virus infection as we did not observe any expression on the virus-free counterpart. The varied expression of *trx* across different isolates and inoculation methods underscores the importance of redox balance and protein reduction processes in oxidative stress management. Furthermore, the observed variability in the expression levels of genes encoding *sod*, *cat*, and *trx* among the four *B. bassiana* isolates during *T. molitor* infection highlights the complex and isolate-specific nature of the fungal response to oxidative stress (Figure 4). These observations suggest that detoxification of immune-related oxidative defenses mediated by these genes occurs during the internal growth phase of the fungus.

In agreement with the literature, the results indicate that heat shock proteins play a crucial role in the survival and stress tolerance of fungi, particularly during the late stages of infection [79,80,83]. Additionally, these findings demonstrate the significance of mannitol metabolism in enhancing the capacity of *B. bassiana* to withstand stress and survive in toxic environments [76].

## 5. Conclusions

This study provides a comprehensive analysis of the symbiotic relationship between mycoviruses and the entomopathogenic fungus *B. bassiana*, with significant implications for biological control applications. Our findings demonstrate that mycovirus replication does not impose a burden on *B. bassiana*. Instead, the mycovirus acts as an extrachromosomal element that enhances the fungus’s virulence and stress tolerance, as evidenced by the comparison between the virus-infected isolate BbVI and its virus-free counterpart BbVF. BbVI exhibited overexpression of key genes associated with virulence and stress response, suggesting that the presence of the mycovirus augments the fungus’s ability to adapt to challenging environments. Genes that participate in carbohydrate assimilation and transportation, Ca^2+^ and iron binding and transportation, secondary metabolite biosynthesis, biotic and abiotic stress tolerance, multidrug resistance efflux pumps, cuticle adherence, cuticle penetration, and nitrate assimilation were overexpressed in BbVI both in vitro and in vivo.

The insect cuticle poses a substantial barrier to infection due to its deficiency in nutrients and water. The enhanced gene expression observed in BbVI may facilitate overcoming these obstacles, thereby increasing pathogenicity. Moreover, the intricate expression patterns of genes involved in calcium and iron binding highlight the fungus’s ability to regulate essential physiological processes in response to intracellular calcium levels and iron availability during infection. These mechanisms are crucial for maintaining cellular homeostasis and facilitating successful host infection. Similarly, the strategic production of mycotoxins and the activation of stress tolerance mechanisms further bolster the fungus’s adaptive response, ensuring its ability to overcome host defenses and environmental challenges during infection.

Additionally, our comparison of inoculation methods revealed distinct gene expression patterns between spraying and injection. The uniform downregulation of certain genes following spraying indicates that these genes are either not required or actively repressed during mid-infection. In contrast, their upregulation following injection underscores their importance during late infection stages. This adaptive regulation reflects *B. bassiana*’s capacity to modulate its physiological processes in response to environmental cues and host interactions. It is noteworthy that while high gene expression levels were detected, they may not directly correspond to protein abundance due to post-transcriptional regulatory mechanisms inherent in eukaryotes. Therefore, further proteomic studies are warranted to elucidate the functional outcomes of these gene expression changes.

To conclude, our findings enhance the understanding of *B. bassiana*’s pathogenicity and adaptation strategies, offering valuable insights into improving the efficacy and stability of this biocontrol agent. By exploring and manipulating these molecular mechanisms, it is possible to optimize *B. bassiana* for more effective use in pest management. This study underscores the potential of mycoviruses as beneficial extrachromosomal genetic elements that can endow entomopathogenic fungi with enhanced virulence and stress tolerance, contributing to more efficient biological control strategies.

## Figures and Tables

**Figure 1 jof-11-00063-f001:**
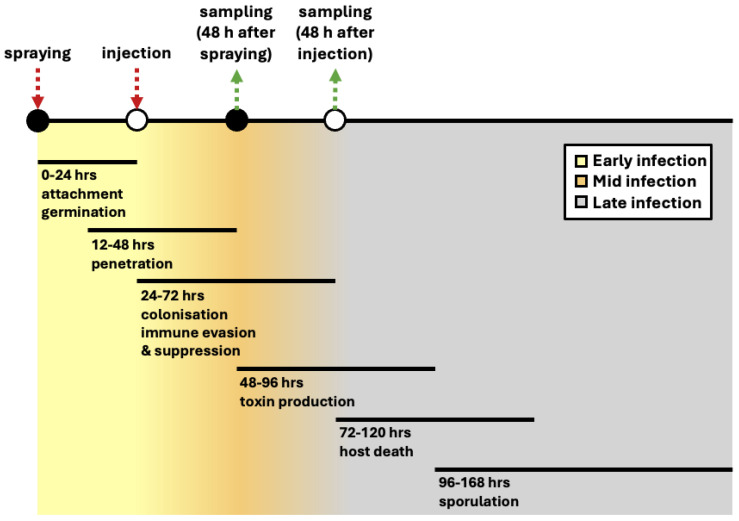
Stages of infection. The timeline is divided into early, mid-, and late infection phases, with the yellow-to-gray gradient reflecting the transition from early to late infection stages. During the early infection phase (0–24 h), the fungus attaches to the cuticle and germinates. In the mid-infection phase (12–72 h), the fungus penetrates and subsequently colonizes the host tissue, while evading and suppressing the insect immune response (24–72 h) and initiating toxin production (48–96 h). In the late infection phase (72–168 h), the fungus kills its host and sporulates. Colored arrows indicate key intervention points, with red representing infection methods, and green representing sampling time points based on infection method.

**Figure 2 jof-11-00063-f002:**
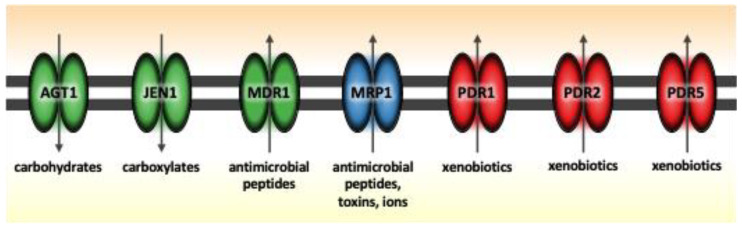
Differential expression of *Beauveria bassiana* transporter genes in vitro. The carboxylic acid transporter gene *jen1*, the carbohydrate transporter gene *agt1* and the multidrug resistance transporter gene *mdr1* (ABC-B) are substantially upregulated in BbVI compared to BbVF, as depicted in green. The multidrug resistance transporter gene *mrp1* (ABC-C) shows similar expression levels in both strains, as depicted in blue, while the ABC-G pleiotropic drug resistance transporter genes *pdr1*, *pdr2*, and *pdr5* are switched off in both strains, as depicted in red. The arrows in the figure indicate the direction of transport: influx arrows represent the intake of carboxylates and carbohydrates, while efflux arrows represent the export of antimicrobial peptides, toxins, ions and xenobiotics (modified from Ortiz-Urquiza et al., 2016) [40].

**Figure 3 jof-11-00063-f003:**
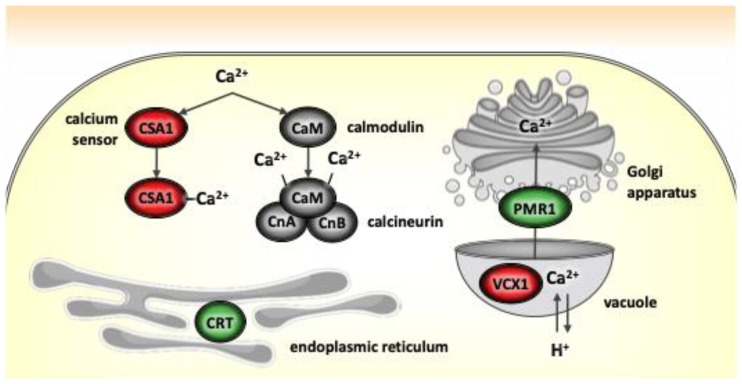
Differential expression of *Beauveria bassiana* genes involved in calcium signaling and transport in vivo following injection. The calreticulin (*crt*) and Golgi calcium transporter *pmr1* genes are upregulated in BbVI compared to BbVF, as depicted in green. Conversely, the calcium sensor acidification *csa1* and vacuolar calcium transporter *vcx1* genes are switched off in both strains, as depicted in red (CaM, calmodulin; CnA/B, calcineurin catalytic and regulatory subunits; modified from Ortiz-Urquiza et al., 2016) [40].

**Figure 4 jof-11-00063-f004:**
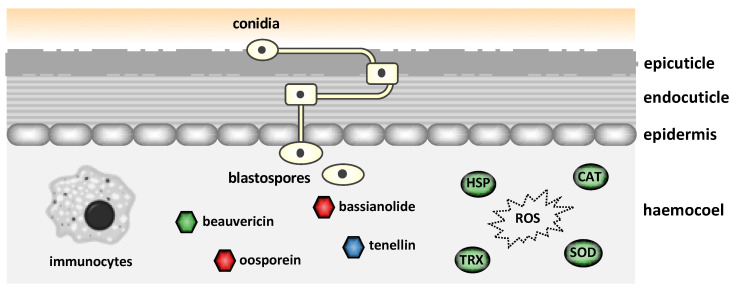
Differential expression of *Beauveria bassiana* genes involved in mycotoxin production and oxidative stress in vivo following injection. Following conidia attachment and penetration of the insect cuticle, the fungus secretes toxins such as beauvericin, bassianolide, oosporein and tenellin. The beauvericin biosynthesis gene (*beas*) is upregulated in BbVI compared to BbVF, depicted in green. The biosynthesis genes for bassianolide (*bsls*) and oosporein (*osp1*) are switched off in both strains, depicted in red. The tenellin biosynthesis gene (*tens*) is expressed at similar levels in both BbVI and BbVF, depicted in blue. Additionally, the genes encoding catalase (*cat*), superoxide dismutase (*sod*), thioredoxin (*trx*), and heat shock proteins (*hsp*) are upregulated in BbVI compared to BbVF, depicted in green.

**Table 1 jof-11-00063-t001:** In vitro relative expression levels and quartile rankings of genes involved in carbon assimilation and transport in BbVF and BbVI. The time points are recorded at days post inoculation (dpi), with symbols indicating upregulation (↑), or same expression (–). The quartile rankings provide a relative measure of gene expression levels within each transcriptome, with (0) indicating undetectable expression.

Gene		Time Point	Expression BbVI vs. BbVF	Quartile		Function
Name	ID			BbVF	BbVI	
*agt1*	853207	7 dpi 21 dpi	↑ ↑	2 0	3 2	Uptake of alpha-glucosides
*jen1*	70112078	10 dpi 14 dpi	– ↑	3 0	3 2	Lactate transporter
*mdr1*	19884037	7 dpi	↑	3	4	Peptide transport
*mrp1*	19886866	7 dpi	–	3	3	Ion transport and/or toxin secretion

**Table 2 jof-11-00063-t002:** In vivo expression levels and quartile rankings for genes involved in carbon assimilation across four different samples: BbVF, BbVI, Naturalis, and BotaniGard, following injection (left) or spraying (right). The quartile rankings provide a relative measure of gene expression levels within each transcriptome, with (0) indicating undetectable expression. The final column describes the function of each gene.

Gene		Quartile								Function
Name	ID	BbVF		BbVI		Naturalis		BotaniGard		
*agt1*	853207	0	0	2	0	0	0	0	0	Uptake of alpha-glucosides
*jen1*	70112078	0	0	0	0	0	0	0	0	Lactate transporter
*mdr1*	19884037	0	0	0	0	0	0	0	0	Peptide transport
*mrp1*	19886866	0	0	3	0	3	0	2	0	Ion transport and/or toxin secretion
*pmd1*	9537001	0	0	2	0	2	0	0	0	Ion transport and/or toxin secretion
*hxtA*	2870633	0	0	2	1	4	0	0	0	Glucose uptake
*rco3*	63742076	0	0	3	0	4	0	0	0	Glucose transport
a-1,3-glucanase	63836283	0	0	3	0	3	0	1	0	Hydrolysis of β-1,3-glucan bonds

**Table 3 jof-11-00063-t003:** In vivo expression levels and quartile rankings for genes involved in calcium and iron binding across four different samples: BbVF, BbVI, Naturalis, and BotaniGard, following spraying (mid-infection, left) or injection (late infection, right). The quartile rankings provide a relative measure of gene expression levels within each transcriptome, with (0) indicating undetectable expression. The final column describes the function of each gene.

Gene		Quartile								Function
Name	ID	BbVF		BbVI		Naturalis		BotaniGard		
*csa1*	30021673	0	0	0	0	0	0	0	0	Calcium-binding protein
*crt*	19892741	0	0	0	4	0	4	0	3	Calcium transporter
*pmr1*	852709	0	0	0	2	0	0	0	0	P-type Ca^2+^ ATPase
*vcx1*	2539393	0	0	0	0	0	0	0	0	Calcium/proton antiporter
*fet3*	18167151	0	0	0	4	0	4	0	2	Iron transport multicopper ferroxidase
*mfs*	19890043	4	4	3	4	3	4	4	4	Major facilitator superfamily

**Table 4 jof-11-00063-t004:** In vitro expression patterns and quartile rankings of genes involved in secondary metabolite biosynthesis in BbVF and BbVI. The time points are recorded at days post inoculation (dpi), with symbols indicating upregulation (↑), or same expression (–). The quartile rankings provide a relative measure of gene expression levels within each transcriptome, with (0) indicating undetectable expression.

Gene		Time Point	Expression BbVI vs. BbVF	Quartile		Function
Name	ID			BbVF	BbVI	
*beas*	19892739	4 dpi 14 dpi 21 dpi	↑ ↑ –	0 0 1	1 1 1	Beauvericin (mycotoxin) synthesis
*bsls*	19885642	7 dpi 21 dpi	– –	0 0	0 0	Bassianolide (mycotoxin) synthesis
*osp1*	VWP00012.1	7 dpi 21 dpi	– –	0 0	0 0	Oosporein (mycotoxin) biosynthesis protein 1
*tens*	59244537	7 dpi 21 dpi	↑ ↑	1 0	2 3	Tenellin (iron chelator) synthesis

**Table 5 jof-11-00063-t005:** In vivo expression levels and quartile rankings for genes involved in secondary metabolite biosynthesis across four different samples: BbVF, BbVI, Naturalis, and BotaniGard, following injection (left) or spraying (right). The quartile rankings provide a relative measure of gene expression levels within each transcriptome, with (0) indicating undetectable expression. The final column describes the function of each gene.

Gene		Quartile								Function
Name	ID	BbVF		BbVI		Naturalis		BotaniGard		
*beas*	19892739	0	0	3	0	1	1	1	1	Beauvericin (mycotoxin) synthesis
*bsls*	19885642	0	0	0	0	0	0	0	0	Bassianolide (mycotoxin) synthesis
*osp1*	VWP00012.1	0	0	0	0	0	0	0	0	Oosporein (mycotoxin) biosynthesis protein 1
*tens*	59244537	4	3	4	3	4	3	4	3	Tenellin (iron chelator) synthesis
*bba*	19892355	0	0	0	0	2	0	0	0	Heat-labile enterotoxin
*tox2*	20369264	0	0	2	1	1	0	0	0	HC-toxin

**Table 6 jof-11-00063-t006:** In vitro expression patterns and quartile rankings of genes involved in biotic and abiotic stress tolerance in BbVF and BbVI. The time points are recorded at days post inoculation (dpi), with symbols indicating upregulation (↑), downregulation (↓), or same expression (–). The quartile rankings provide a relative measure of gene expression levels within each transcriptome, with (0) indicating undetectable expression.

Gene		Time Point	Expression BbVI vs. BbVF	Quartile		Function
Name	ID			BbVF	BbVI	
*hsp30*	19893220	4 dpi 14 dpi	– –	4 4	4 4	Conidial thermotolerance
*hsp70*	19888598	4 dpi 14 dpi 21 dpi	– ↑ ↑	4 3 3	4 4 4	Growth, conidiation and cell wall integrity
*hsp90*	18169849	14 dpi 21 dpi	– –	4 4	4 4	MAPK pathway activation
*mpd*	19885153	4 dpi 14 dpi	– –	4 4	4 4	Mannitol biosynthesis
*mtd*	19889641	10 dpi	–	4	4	Mannitol biosynthesis
*tps1*	19884544	7 dpi 14 dpi	– –	3 3	3 3	Trehalose biosynthesis
*tps2*	855303	7 dpi 14 dpi	↑ ↑	3 3	4 4	Trehalose biosynthesis
*agdC*	77793718	4 dpi 7 dpi 21 dpi	↓ ↓ ↓	4 4 4	2 2 3	Carbohydrate metabolism

**Table 7 jof-11-00063-t007:** In vivo expression levels and quartile rankings for genes involved in biotic and abiotic stress tolerance across four different samples: BbVF, BbVI, Naturalis, and BotaniGard, following spraying (mid-infection, left) or injection (late infection, right). The quartile rankings provide a relative measure of gene expression levels within each transcriptome, with (0) indicating undetectable expression. The final column describes the function of each gene.

Gene		Quartile								Function
Name	ID	BbVF		BbVI		Naturalis		BotaniGard		
*cat*	19892772	0	0	0	4	0	4	0	0	Antioxidant enzyme
*sod*	19887329	0	0	0	3	0	0	0	0	Antioxidant enzyme
*trx*	19892773	0	0	0	4	0	4	2	3	Antioxidant enzyme
*hsp/sti1*	854192	0	0	0	4	0	4	0	0	Stress response–protein folding
*hsp10*	854185	0	0	0	4	0	4	0	0	Stress response–protein folding
*hsp30*	19893220	0	0	0	0	0	0	0	0	Conidial thermotolerance
*hsp70*	19888598	0	0	2	4	0	4	0	2	Growth, conidiation and cell wall integrity
*hsp90*	18169849	0	0	0	4	0	2	0	2	MAPK pathway activation
*mpd*	19885153	0	0	0	2	0	0	0	0	Mannitol biosynthesis
*mtd*	19889641	0	0	0	0	0	0	0	0	Mannitol biosynthesis
*tps1*	19884544	0	0	0	3	0	0	0	0	Trehalose biosynthesis
*tps2*	855303	0	0	0	0	0	0	0	0	Trehalose biosynthesis

## Data Availability

The raw data supporting the conclusions of this article will be made available by the authors on request.

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
