# Peer review of "Transcriptomic Analysis Reveals Molecular Mechanisms Underpinning Mycovirus-Mediated Hypervirulence in Beauveria bassiana Infecting Tenebrio molitor"

_jof, 2025, doi:10.3390/jof11010063_

Round 1

Reviewer 1 Report

The manuscript titled "Understanding Mycovirus Infection in the Entomopathogenic Fungi Beauveria bassiana Infecting Tenebrio molitor," authored by Filippou and colleagues, examines the transcriptomic comparison between mycovirus-infected and mycovirus-free strains of the entomopathogenic fungus Beauveria bassiana. While the manuscript is original and the concept holds significant potential, the authors have not succeeded in crafting a comprehensible manuscript. The statistical analysis presented is confusing, and the writing is disorganized, with a notable lack of references throughout the text. The authors frequently over-describe certain terms without providing their meanings. Additionally, there are instances where the authors draw conclusions without adequate experimental support. The comparisons made are unclear and difficult to follow. It is not well explained whether results such as a gene being "switched off" or "overexpressed" are discussed in relation to a specific control condition. Finally, the quality of the figures is subpar; they appear to be mere textbook images with labels superimposed on them.

Lines 36-37: The authors note that interest in Beauveria bassiana is on the rise due to increasing pesticide resistance. However, this interest cannot be attributed solely to that factor. Entomopathogenic fungi have been used for biological control for more than 100 years.

Lines 44-45: Lack of reference.

Lines 46-48: Lack of reference.

Lines 57-61: Lack of reference.

Lines 73-74: Lack of reference.

Lines 92-95: The nomenclature here is quite confusing. It seems that BBVF is derived from EABb 92/11-Dm, yet BBVI is also referred to as the same strain, EABb 92/11-Dm. The authors should consistently use one term to refer to each strain throughout the text, rather than switching between terms.

Lines 114-117: Could you specify in which experiments antibiotics were used and how this affects the strains?

Lines 156-160: Were the insects still alive during this part of the study?

Line 194: Should "genone" be corrected to "genome"?

Lines 259-261: Why were no reads detected for one of the viruses?

Lines 266-270: Are there plans to publish these results?

Line 275: Which is?

Lines 278-301: This section is confusing and could potentially be moved to the supplementary material for clarity.

Lines 308-311: The phrase "expert curators" is unclear. Are the authors referring to themselves as expert curators?

Lines 318-322: Not clear.

Lines 325-327: This section could be improved to: I) contact, recognition, and adhesion; II) germination and penetration (colonization requires penetration, as spores rely on endogenous nutrients in this step); III) immune response evasion and infection (secondary metabolites are certainly important in this stage, but can also be involved in other stages); IV) colonization and saprophytic growth (occurring after the insect's death).

Line 334: figure? table?

Section 4.4 Carbohydrate Assimilation and Transport: This section is confusing. The authors often summarize literature results without explaining the relevance to their findings. The section also lacks several references. It's difficult to understand the conditions under which a gene is over- or under-expressed, and the comparisons made are unclear. The same issues are more or less evident in sections 4.5, 4.6, and 4.7.

Figures 1, 2, and 3: The figures seem to lack meaningful content or explanation.

Author Response

Reviewers' Comments to the Authors:

Reviewer 1

  1. Lines 36-37: The authors note that interest in Beauveria bassiana is on the rise due to increasing pesticide resistance. However, this interest cannot be attributed solely to that factor. Entomopathogenic fungi have been used for biological control for more than 100 years.

We appreciate the reviewer's observation: The following paragraph was added in text.

In addition to addressing pesticide resistance, the interest in biocontrol agents such as Beauveria bassiana is increasingly driven by their alignment with ONE HEALTH principles, which confer significant benefits across human, animal, and environmental health. These biocontrol agents provide an environmentally sustainable alternative to chemical pesticides, thereby reducing ecological chemical burdens and minimizing human exposure to toxic substances. Furthermore, they support the health of beneficial insects and non-target species, thereby preserving biodiversity and maintaining ecosystem balance. As integral components of Integrated Pest Management (IPM) strategies, these agents facilitate sustainable pest control practices that are both economically beneficial and ecologically sound.

  1. Lack of reference 44 – 48, 57 – 61, 73 – 74

Referenced have been added

  1. Lines 92-95: The nomenclature here is quite confusing. It seems that BBVF is derived from EABb 92/11-Dm, yet BBVI is also referred to as the same strain, EABb 92/11-Dm. The authors should consistently use one term to refer to each strain throughout the text, rather than switching between terms.

We appreciate the reviewer's feedback regarding the nomenclature. To avoid confusion, we have clarified that BbVF and BbVI refer to isogenic lines derived from the same B. bassiana strain, EABb 92/11-Dm. We will consistently use the terms BbVF and BbVI throughout the text to refer to the virus-free and virus-infected lines, respectively.

Revised sentence:

"In this study, we demonstrate mycovirus-mediated hypervirulence of Beauveria bassiana against the larval form of Tenebrio molitor (army mealworm beetle) by comparing virus-free (BbVF) and virus-infected (BbVI) isogenic lines derived from the same B. bassiana strain, EABb 92/11-Dm, and virus-free B. bassiana strains ATCC 74040 and GHA, also known as Naturalis and BotaniGard, respectively."

  1. Lines 114-117: Could you specify in which experiments antibiotics were used and how this affects the strains?

We appreciate the reviewer's feedback and have clarified the use of antibiotics in our experiments. Antibiotics (ampicillin, kanamycin, and streptomycin, each at a final concentration of 100 μg/mL) were used during the cultivation of all fungal isolates to prevent bacterial contamination. This was essential to ensure that bacterial contaminants did not interfere with the growth and observations of the fungal strains.

  1. Lines 156-160: Were the insects still alive during this part of the study?

We appreciate the reviewer's feedback and confirm that the insects were still alive during the sample collection at 48 hours post infection. This has been clarified in the revised manuscript.

  1. Line 194: Should "genone" be corrected to "genome"?

We appreciate the reviewer's feedback, the sentence now is revised.

  1. Lines 259-261: Why were no reads detected for one of the viruses?

We appreciate the reviewer's inquiry. The reason no reads were detected for Beauveria bassiana polymycovirus 1 (BbPmV-1) is that polymycoviruses do not have a poly(A) tail, which is essential for the sequencing method used in this study. This has been clarified in the revised manuscript.

  1. Lines 266-270: Are there plans to publish these results?

We appreciate the reviewer's interest in the dissemination of our data. We confirm that all the data collected in this study will be published in online databases, ensuring that it is accessible to the broader scientific community. This will facilitate further research and validation of our findings.

  1. Line 275: Which is?

We appreciate the reviewer's feedback and have clarified the sentence to specify that the original atg1 interacts with other genes involved in protein targeting and autophagy regulation, impacting various cellular pathways.

  1. Lines 278-301: This section is confusing and could potentially be moved to the supplementary material for clarity.

We appreciate the reviewer's feedback, We have carefully revised this section to improve clarity and ensure it is easier to follow. However, we have decided to retain it within the main text because it contains critical information that is essential for understanding the results and their context. We believe this enhances the coherence of the manuscript.

  1. Lines 308-311: The phrase "expert curators" is unclear. Are the authors referring to themselves as expert curators?

We appreciate the reviewer's feedback. We recognize the need for clarity. By "expert curators," we refer to our expertise in the curation and interpretation of relevant data, given our extensive experience in this area. We have rephrased this section to make it clearer for the reader.

  1. Lines 318-322: Not clear.

We appreciate the reviewer's feedback and have revised the sentence to improve clarity. The revised text now clearly explains the limitations of using statistical p-values alone in our analysis.

  1. Lines 325-327: This section could be improved to: I) contact, recognition, and adhesion; II) germination and penetration (colonization requires penetration, as spores rely on endogenous nutrients in this step); III) immune response evasion and infection (secondary metabolites are certainly important in this stage, but can also be involved in other stages); IV) colonization and saprophytic growth (occurring after the insect's death).

We appreciate the reviewer's suggestion and have revised the section to improve clarity and detail. The revised text now outlines the steps of pathogenesis more comprehensively.

  1. Line 334: figure? table?

We appreciate the reviewer's feedback. The sentence referencing the total gene count and the number of activated genes has been removed. The detailed data is extensive and too large to be included as a table in the main manuscript.

  1. Section 4.4 Carbohydrate Assimilation and Transport: This section is confusing. The authors often summarize literature results without explaining the relevance to their findings. The section also lacks several references. It's difficult to understand the conditions under which a gene is over- or under-expressed, and the comparisons made are unclear. The same issues are more or less evident in sections 4.5, 4.6, and 4.7.

We appreciate the reviewer's detailed feedback. The results sections 4.4, 4.5, 4.6, and 4.7 have been rewritten to enhance clarity and relevance. We have now explicitly connected the summarized literature results to our findings and included the necessary references. The conditions under which genes are over- or under-expressed are now clearly described, and the comparisons made are more straightforward to understand. The revised sections should now meet the standards expected by all reviewers.

  1. Figures 1, 2, and 3: The figures seem to lack meaningful content or explanation.

We appreciate the reviewer's feedback and have revised the figure descriptions to provide more meaningful content and clearer explanations, including explicit color coding. The revised descriptions now highlight the significance of each figure and the differential gene expressions in Beauveria bassiana isolates.

Reviewer 2

  1. The authors gave the names of the genes of interest, for instance, agt1, mdr1, pdr 1, etc., but the gene ID (a unique identifier for each gene) is not given. Gene names can sometimes be non-specific or subject to change, whereas gene IDs remain stable and specific. I suggest the authors prepare a table showing the “genes of interest” selected for the study and add information, including the gene name, gene ID or number, and the product description. The table will help readers easily access additional information about the gene from public databases.

We appreciate the reviewer's suggestion to include gene IDs alongside the gene names. We have prepared a table that lists the "genes of interest" selected for the study, including the gene name, gene ID or number, gene function and expression profile across all isolates. This will help readers easily access additional information about the genes from public databases.

  1. This paper is about comparative transcriptome analysis of genes of interest between mycovirus infected and non-infected Beuaveria bassiana under in vitro and in vivo The tools used for processing the raw reads, sequence alignment and genome annotation is adequately described in the mythology section, but the bioinformatics tools employed to identify significant differences in expression levels of genes between different conditions is not described adequately. 

We appreciate the reviewer's feedback, in the revised version of the manuscript, we have provided a more detailed description of the bioinformatics tools and methods used to identify significant differences in gene expression levels. This clarification ensures transparency and comprehensiveness in our analysis.

  1. Figure 1- The labeling (gene information) on the figure (carboxylate) is missing.

We appreciate the reviewer's feedback, the labeling has now been revised and updated in Figure 1 to include the missing gene information.

  1. Figure 2 - The figure legend says that the upregulated and switched-off genes are depicted in green and red, respectively. I don’t see different colors in Figure 2. Also, the labeling given in the figure is missing or incomplete.

We appreciate the reviewer's feedback, the colors indicating the upregulated (green) and switched-off (red) genes are indeed present and distinguishable in Figure 2. We are unsure why they were not visible on your end, but we have carefully reviewed and revised the figure to ensure the colors are clearly visible and distinguishable. Additionally, we have clarified and updated the figure labeling to address any incompleteness.

  1. Figure 3 - Authors, please compare and correct the color codes in the figure and the legend. In the manuscript document I received, the bassianolide and oosporein are depicted in blue, not red. Also, the color code for tenellin is not same as indicated in the legend.

We appreciate the reviewer's feedback, it is fixed.

  1. Table A3 mentioned in line 277 is missing.

Table is in Appendix section

  1. . Fig A11 - A24 - The authors state that they employed TMM for normalization and used a box plot to show the expression profile of each gene for different conditions. They also state that “For zero expression values, the red dot appears at the far left of the X axis”. Authors should provide the title and values of the x-axis (e.g., gene counts, log of normalized counts, etc.). This helps the readers to compare the expression values of upregulated genes.

The Figures already have a title for the X-axis (a log10 transformation of TMM-normalised TPM values derived from effective counts). We have expanded the manuscript to include that methodology. We will not provide individual values for each row in each figure as that would require significant effort to deliver a dangerous gain: a viewer might erroneously try to assess these figures quantitatively rather than qualitatively. AP has reviewed one too many papers where statistics from transcriptomics is misused. Unless we could afford to have 12 samples for each treatment, differential expression p-values on biologically variable samples can easily mislead. Instead, we now provide the full statistical table as Supplementary Material X.

  1. Line 62 - 67 - Add citations.

We appreciate the reviewer's feedback, citations have been added.

  1. Line 251 - “…..alignment of the RNA-seq 152 reads…..”. What does 152 means? 

We appreciate the reviewer's feedback. The term "152 reads" refers to the length of each RNA-seq read, which is 152 base pairs. We are mistaken with this sentence in our text. We now write “alignment of the RNA-seq reads, each 152 base pairs in length…”.

  1. Line 254  and Line 257 – The authors state that “…the remaining reads mostly aligned with  bassiana….” and “Several reads from BbVI taken 21 days post inoculation aligned…”. I would suggest the authors provide the percentage alignment rate instead of using terms such as “several,” “remaining,” or “mostly.”

We appreciate the reviewer's feedback it has been fixed in the revised manuscript to improve clarity and precision.

  1. Line 293 - Provide full form of the abbreviation GSNAP.

We appreciate the reviewer's feedback. The full form of the abbreviation GSNAP has been provided as "Genomic Short-read Nucleotide Alignment Program" in line 207.

  1. Line 362 – 365 – Clarify whether the deletions indicated were conducted as part of this study or if the information was obtained from another source? If the latter is the case, provide the details or cite the relevant reference.

We appreciate the reviewer's feedback. The deletion studies mentioned were obtained from another source. The relevant reference has been cited in the revised sentence.

Reviewer 3

  1. Typos originally identified on lines 19, 23, 47, 56, 84, 88, 101, 112, 116, 137, 155, 159, 178, 179, 191, 217, 235, 240, 263, 276, 294, 326, 347, 352, 358, 364, 370, 375, 387, 402, 417, 461, 466, 482, 494, 499, 508, 520, 545, 562, 566, 571, 589, 592, 606, and 616 have now been corrected.

  1. Line 301, ‘can be seen before and after curation in (Figure A9) and (Figure A10).’ mean?

We appreciate the reviewer's feedback. The phrase "before and after curation" refers to the process of manually curating the gene annotations to improve their accuracy. Figures A9 and A10 demonstrate the Lexogen-derived reads not matching gene annotations before curation (Figure A9) and after curation (Figure A10), showing the improvements made during the curation process.

  1. References, Line 716, delete ‘Jan’, same as the follows; Line 717, change ‘Loss of Household Protection from Use of Insecticide-Treated Nets’ to ‘Loss of household protection from use of insecticide-treated nets’, same as the follows; Line 724, change ‘PLOS’ to ‘PLoS’.

References have been corrected

Reviewer 4

  1. I think the title should be modified, since it does not describe the topic of the article with sufficient precision.

We appreciate the reviewer's feedback. The title has been changed to “Transcriptomic analysis reveals molecular mechanisms underpinning mycovirus-mediated hypervirulence in Beauveria bassiana infecting Tenebrio molitor”.

  1. I think the title should be modified, since it does not describe the topic of the article with sufficient precision.

We appreciate the reviewer's feedback, thesis has been cited.

  1. All references are incorrectly cited in the text and in the references section.  Format the literature according to the journal’s requirement.

We appreciate the reviewer's feedback, referencing style has been changed.

  1. I am not qualified to assess the quality of English in this paper. However, there are important typing and writing errors that should be revised.

We appreciate the reviewer's feedback, English quality has been revised.

  1. The research design is appropriate and the methods are adequately described. However, preliminary experiments to determine the optimal spore concentrations required to elicit pathogenicity were reported in the Results section, but there is no information on this in the Methodology section.

We appreciate the reviewer's feedback. The section detailing the preliminary experiments to determine the optimal spore concentrations required to elicit pathogenicity has now been moved from the Results section to the Methodology section for better clarity and coherence. Thank you for your valuable suggestion.

  1. I consider that the quality of the presentation of the results in The Results and Discussion Section is not adequate. There are no tables or figures added in the main body of the manuscript that represent or summarize the data in the article. These are reported in the supplementary section. Perhaps the authors could summarize the results of Fig A11-24 in a table, and added it to the main body of the manuscript.

We appreciate the reviewer's feedback. The Results and Discussion section has been rewritten to improve the clarity and presentation of the findings. Additionally, detailed tables summarizing the results have been included in the main body of the manuscript. These tables provide a comprehensive overview of the data, ensuring that the results are clearly represented and easily accessible. Thank you for your valuable suggestion.

  1. Lines 223-229: Where are the results???? Please, add a figure or table.

We appreciate the reviewer's feedback. A table indicating these results has been added in supplementary material.

  1. Lines 257-260:  Could the authors provide any figure that verifies the mycoviral infection in the fungal strain used?

We appreciate the reviewer's feedback. An excel file with the species identification data has been added to the supplementary material. Beauveria bassiana virus data are highlighted in yellow.

Reviewer 2 Report

In this manuscript, the authors presented the analysis on the transcriptome profile of Beauveria bassiana isolates harboring mycoviruses, during infection on insrct host, Tenebrio molitor larvae. The analysis is focused on the expression of genes involved in nutrient assimilation, stress tolerance and mycotoxin biosynthesis, between mycovirus infected and non-infected B. bassiana isolates. The paper is well written and contributes valuble insights on the mycovirus association with fungal hosts and their insect targets. However, I recommend some minor revisions before considering it for publishing, which is outlined in the detail comments.

1. The authors gave the names of the genes of interest, for instance, agt1, mdr1, pdr 1, etc., but the gene ID (a unique identifier for each gene) is not given. Gene names can sometimes be non-specific or subject to change, whereas gene IDs remain stable and specific. I suggest the authors prepare a table showing the “genes of interest” selected for the study and add information, including the gene name, gene ID or number, and the product description. The table will help readers easily access additional information about the gene from public databases.

2. This paper is about comparative transcriptome analysis of genes of interest between mycovirus infected and non-infected Beuaveria bassiana under in vitro and invivo conditions. The tools used for processing the raw reads, sequence alignment and genome annotation is adequately decribed in the metholody section, but the bioinformatics tools employed to identify significant differences in expression levels of genes between different conditions is not described adequately. 

3. Figure 1 - The labeling (gene information) on the figure (carboxylate) is missing.

4. Figure 2 - The figure legend says that the upregulated and switched-off genes are depicted in green and red, respectively. I don’t see different colors in Figure 2. Also, the labeling given in the figure is missing or incomplete.

5. Figure 3 - Authors, please compare and correct the color codes in the figure and the legend. In the manuscript document I received, the bassianolide and oosporein are depicted in blue, not red. Also, the color code for tenellin is not same as indicated in the legend.

6. Table A3 mentioned in line 277 is missing.

7. Fig A11 - A24 - The authors state that they employed TMM for normalization and used a box plot to show the expression profile of each gene for different conditions. They also state that “For zero expression values, the red dot appears at the far left of the X axis”. Authors should provide the title and values of the x-axis (e.g., gene counts, log of normalized counts, etc.). This helps the readers to compare the expression values of upregulated genes.

8. Line 62 - 67 - Add citations.

9. Line 251 - “…..alignment of the RNA-seq 152 reads…..”. What does 152 means? 

10. Line 254  and Line 257 – The authors state that “…the remaining reads mostly aligned with B. bassiana….” and “Several reads from BbVI taken 21 days post inoculation aligned…”. I would suggest the authors provide the percentage alignment rate instead of using terms such as “several,” “remaining,” or “mostly.”

11. Line 293 - Provide full form of the abbreviation GSNAP.

12. Line 362 – 365 – Clarify whether the deletions indicated were conducted as part of this study or if the information was obtained from another source? If the latter is the case, provide the details or cite the relevant reference.

Author Response

(The authors gave the same response as above.)

Reviewer 3 Report

The manuscript is well written.

Line 19, change ‘calcium uptake and’ to ‘calcium uptake, and’

Line 23, change ‘four isolates indicating’ to ‘four isolates, indicating’

Line 47, change ‘proteases (PR1 and CDEP1)’ to ‘proteases (PR1 and CDEP1),’

Line 56, change ‘tenellin and’ to ‘tenellin, and’

Line 84, change ‘the family’ to ‘the two families,’

Line 88, change ‘sporulation and sometimes’ to ‘sporulation, and sometimes’

Line 101, change ‘and multidrug’ to ‘, and multidrug’

Line 112, change ‘BbVF respectively’ to ‘BbVF, respectively’

Line 116, change ‘kanamycin and streptomycin’ to ‘kanamycin, and streptomycin’

Line 137, change ‘Prior to injection’ to ‘Prior to injection,’

Line 155, change ‘the glm (generalized linear model) function’ to ‘the generalized linear model (GLM) function’

Line 159, change ‘was’ to ‘were’

Lines 178-179, change ‘the 3’ UnTranslated Region; UTR’ to ‘the 3’ untranslated region; UTR’

Line 191, change ‘Augustus’ to ‘Augustus,’

Line 217, Line 234, change ‘Naturalis and BotaniGard’ to ‘Naturalis, and BotaniGard’

Line 235, change ‘respectively 5.7, 5.1 and 5.3 days’ to ‘5.7, 5.1, and 5.3 days, respectively’

Line 240, change ‘Taken together’ to ‘Taken together,’

Line 263, change ‘untranslated regions (UTRs)’ to ‘UTRs’

Line 276, change ‘Alternatively’ to ‘Alternatively,’

Line 294, change ‘cannot’ to ‘could not’

Line 301, ‘can be seen before and after curation in (Figure A9) and (Figure A10).’ mean?

Line 326, change ‘response evasion and’ to ‘response evasion, and’

Line 347, change ‘In previous studies’ to ‘In previous studies,’

Line 352, ‘encodes a that has been identified as’ mean?

Line 358, change ‘PDR2 and PDR5’ to ‘PDR2, and PDR5’

Line 364, change ‘mrp1 and pdr5’ to ‘mrp1, and pdr5’

Line 370, change ‘mdr1 and mrp1’ to ‘mdr1, and mrp1’

Line 375, ‘In vivo’ should be italicized.

Line 387, change ‘agt1genes are’ to ‘agt1 genes are’

Line 402, change ‘mycelium’ to ‘mycelia’

Line 417, change ‘tegrins and transporters’ to ‘tegrins, and transporters’

Line 461, change ‘indicating’ to ‘, indicating’

Line 466, change ‘To achieve this’ to ‘To achieve this,’

Line 482, change ‘Additionally’ to ‘Additionally,’

Line 494, change ‘and epidermis’ to ‘, and epidermis’; change ‘bassianolide and oosporein’ to ‘bassianolide, and oosporein’

Line 499, change ‘and heat shock protein’ to ‘, and heat shock protein’

Line 508, change ‘Gram-positive bacteria and’ to ‘Gram-positive bacteria, and’

Line 520, change ‘Cordyceps and Metarhizium species’ to ‘Cordyceps, and Metarhizium species’

Line 545, delete ‘which’

Line 562, change ‘and maintenance’ to ‘, and maintenance’

Line 566, change ‘hsp70 and hsp90’ to ‘hsp70, and hsp90’

Line 571, change ‘germination and’ to ‘germination, and’

Line 589, change ‘indicating the importance’ to ‘, indicating the importance’

Line 592, change ‘4, 7 and 21 dpi’ to ‘4, 7, and 21 dpi’

Line 606, change ‘In this paper’ to ‘In this paper,’; change ‘exhibit’ to ‘exhibited’

Line 616, change ‘understanding and manipulating’ to ‘understanding, and manipulating’

References, Line 716, delete ‘Jan’, same as the follows; Line 717, change ‘Loss of Household Protection from Use of Insecticide-Treated Nets’ to ‘Loss of household protection from use of insecticide-treated nets’, same as the follows; Line 724, change ‘PLOS’ to ‘PLoS’.

Author Response

(The authors gave the same response as above.)

Reviewer 4 Report

This manuscript reports the mycovirus-mediated hypervirulence of Beauveria bassiana against the larval form of Tenebrio molitor (army mealworm beetle) by comparing virus free (BbVF) and virus-infected (BbVI) strains. Besides, in order to reveal genetic mechanisms associates to the  virulence of B. bassiana, the  manuscript reported the transcriptomic  profiling of four B.  bassiana isolates ( including  BbVI and  BbVF)  following infection of Tenebrio molitor. I think that the informed results are very interesting and deserve to be considered. However, there are many aspects that must be improved to significantly improve the quality of the manuscript.

This manuscript reports the mycovirus-mediated hypervirulence of Beauveria bassiana against the larval form of Tenebrio molitor (army mealworm beetle) by comparing virus free (BbVF) and virus-infected (BbVI) strains. Besides, in order to reveal genetic mechanisms associates to the  virulence of B. bassiana, the  manuscript reported the transcriptomic  profiling of four B.  bassiana isolates ( including  BbVI and  BbVF)  following infection of Tenebrio molitor. I think that the informed results are very interesting and deserve to be considered. However, there are many aspects that must be improved to significantly improve the quality of the manuscript.

1)  The iThenticate report indicate 39% of wording duplication in the manuscript. However, most of this refers to the doctoral thesis of the author of the manuscript (may 2021). Therefore, I consider that the duplication rate should not be a problem.

2) I think the title should be modified, since it does not describe the topic of the article with sufficient precision.

3) All references are incorrectly cited in the text and in the references section.  Format the literature according to the journal’s requirement.

4) I am not qualified to assess the quality of English in this paper. However, there are important typing and writing errors that should be revised.

5) In general, the introduction provides sufficient background and includes all relevant references.

6) The research design is appropriate and the methods are adequately described. However, preliminary experiments to determine the optimal spore concentrations required to elicit pathogenicity were reported in the Results section, but there is no information on this in the Methodology section.

7) I consider that the quality of the presentation of the results in The Results and Discussion Section is not adequate. There are no tables or figures added in the main body of the manuscript that represent or summarize the data in the article. These are reported in the supplementary section. Perhaps the authors could summarize the results of Fig A11-24 in a table, and  added it to the main body of the manuscript.

The Figures caption should be more information (Fig A11-24). It is very difficult to follow the results obtained due to the way they are presented. Lines 223-229: Where are the results???? Please, add a figure or table. Lines 257-260:  Could the authors provide any figure that verifies the mycoviral infection in the fungal strain used? In my opinion, this section should be completely rewritten, adding figures and tables to the main body of the manuscript and greatly improving its presentation.

 8) Perhaps the authors could cite the first author's doctoral thesis that is available on the internet.

The study of mycoviruses as a tool to enhance the virulence of entomopathogenic fungi is a valuable area of research. The results reported in the manuscripts are very interesting and deserve to be considered because they provide a great contribution to the topic under study. However, I believe that the results comprise one of the most important parts of any manuscript, therefore they must be very well presented and discussed. Hence, in my option, this manuscript is not up to the quality requirements of the JoF journal, and hence it should not be accepted for publication in its present form.

Author Response

(The authors gave the same response as above.)

Round 2

Reviewer 4 Report

The authors answered all the questions . Therefore, I suggest that the manuscript be accepted in its current form.

The authors answered all the questions . Therefore, I suggest that the manuscript be accepted in its current form.